# Mechanical Properties and Fatigue Life Analysis of Motion Cables in Sensors under Cyclic Loading

**DOI:** 10.3390/s24041109

**Published:** 2024-02-08

**Authors:** Weizhe Liang, Wei Guan, Ying Ding, Chunjin Hang, Yan Zhou, Xiaojing Zou, Shenghai Yue

**Affiliations:** 1Department of Astronautics and Mechanics, Harbin Institute of Technology, Harbin 150001, China; 15946221149@163.com (W.L.); zyan@stu.hit.edu.cn (Y.Z.); wszxj1996@163.com (X.Z.); 13292187005@163.com (S.Y.); 2China Academy of Space Technology, Beijing Institute of Control Engineering, Beijing 100094, China; 3Welding System, Harbin Institute of Technology, Harbin 150001, China; hangcj@hit.edu.cn

**Keywords:** motion cables, cyclic load, mechanical property, fatigue life

## Abstract

Motion cables, which are widely used in aero-engine sensors, are critical components that determine sensor stability. Because motion cables have unique motion characteristics, the study of their mechanical properties and reliability is very important. In addition, motion cables are complex in structure and cannot be applied to conventional fixed cable research methods. In this study, a new approach is proposed to introduce the theory of anisotropic composites into a simplified cable model, so that the cable is both physically conditioned and has good mechanical properties. While applying the theory of anisotropic composites, the forces of tension and torsion are considered in a motion cable under the combined action. In this context, the reliability of the structure is the fatigue life of the cable. In this paper, the mechanical properties and fatigue life of motion cables are investigated using the finite element method at different inclination angles and fixation points. The simulation results show that there is a positive correlation between the inclination angle and the extreme stress in the motion cables, and the optimal inclination angle of 0° is determined. The number of fixing points should be reduced to minimize the additional moments generated during the movement and to ensure proper movement of the cables. The optimal configuration is a 0° inclination angle and two fixing points. Subsequently, the fatigue life under these optimal conditions is analyzed. The results show that the high-stress zone corresponds to the location of the short-fatigue life, which is the middle of the motion cables. Therefore, minimizing the inclination angle and the number of fixing points of the motion cables may increase their fatigue life and thus provide recommendations for optimizing their reliability.

## 1. Introduction

Motion cables for engine sensors are used in a wide range of applications and are known for their lightweight construction, high conductivity, and high reliability [1]. Currently, there is a lack of comprehensive research on the challenges and reliability issues faced by motion cables in actual operation. Therefore, it is crucial to systematically unveil the performance characteristics of motion cables in engine sensors [2]. Reliability is defined as the ability of a product to fulfill a task under specified conditions and within a specified time. Factors affecting reliability are material quality, environmental conditions, failure rate, and fatigue life, of which fatigue life is one of the key factors affecting reliability [3]. This paper aims to determine the optimal working conditions of motion cables under a variety of working conditions and then analyze the fatigue life of motion cables under the optimal working conditions. Motion cables play a vital role in engine sensors as key sensor connection elements. In sensor applications, these cables must not only meet electrical connection requirements but also adapt to various complex motion environments during engine operation. Motion cables are a special category that move in synchronization with the connecting parts during the use of the product among the various types of cables [4]. Presently, the process design of motion cables faces challenges related to poor wiring process reliability and cable bending fatigue. In addition, the reliability of cables changes with a change in cable structure, which affects the fatigue life of cables. Therefore, it is crucial to study the process and fatigue life of the motion cable assembly.

Giglio [5] delved into the distinctions between motion cables and other structures, highlighting the variance in fatigue life between non-motion structures and motion cables, arising from the special requirements of motion cables within a mechanism. Non-motion structures endure static loads, focusing solely on the ultimate loads they can withstand, without the need to consider fatigue life factors. In contrast, motion cables undergo bending and stretching during motion, requiring millions of cycles of use [6]. This places exceptionally high demands on the fatigue life of the cable material. Consequently, sports cables exhibit significant differences in fatigue life compared to other structures. In response to these challenges, researchers have proposed a series of mathematical models for calculating the mechanical properties of cables [7]. Initially, the mechanical properties of cables are explored through experimental methods. However, due to the high-strength properties of the cables, experimental methods would entail lengthy calculation cycles and diminish the efficiency of cable optimization [8]. Therefore, many researchers have used finite element methods to obtain the mechanical properties of cables [9,10].

Judge [11] analyzed the mechanical properties of multilayer spiral cables under quasi-static axial loads using LS-DYNA R11.0.0. Moreover, studies indicate that even slight bends can significantly impact stress distribution in axial cables. Hence, studying the bending performance of cables becomes imperative. Yu [12] developed a 3D finite element model of a seven-wire steel strand to investigate the transverse loading effects on local stress distribution and life performance. Nawrocki and Labrosse [13] developed a specific model considering the motion of inner wires and examined the relative motion between them. Ghoreishi [14] conducted three-dimensional finite element modeling of “6 + 1” steel wire single-layer strands under static axial loads, comparing it with nine theoretical models of wire elasticity. Jiang [15] performed a concise finite element analysis of seven-wire and three-layer straight steel wire ropes, considering friction and contact effects. They obtained the elastic–plastic properties under tensile and cross-sectional stress distributions. The stress studies in previous cable analyses, focusing on single axial tensile or torsional loading effects, do not apply to the analysis of motion cables on engine sensors. Therefore, proposing a method to analyze the mechanical properties of cables during their motion becomes crucial. However, the above research has focused on the performance of traditional cables, and with the iterative optimization of electronics, motion cables have become the primary connection method in sensors [16]. Since motion cables need to be cycled tens of thousands of times during movement, the study of their fatigue life in service is particularly important. Key factors and parameters need to be considered such as failure data, life distribution, and environmental factors, which directly affect the accuracy of the model when establishing a fatigue life model [17]. Reliability modeling under failure data needs to be based on historical failure data of the system or equipment [18]. This includes information such as when the failure occurred, the type, the cause, and the time required for repair. Lifetime distribution describes the distribution of the lifetime of a system or a component over a certain time frame [19]. Common life distributions include exponential distribution, Weibull distribution, etc. Environmental factors are key factors in fatigue life modeling, and different operating environments have an impact on system reliability. Environmental factors such as temperature, humidity, vibration, etc., are considered to more accurately assess the reliability of the system under different operating conditions [20].

The traditional method of improving cable fatigue life is through experimental testing to determine the factors that reduce cable fatigue life. The method of load experiments is used to study in detail the factors that affect the fatigue life of cables [21]. Load experiments are an effective means of revealing key characteristics of the fatigue performance of cable materials by conducting experiments at different load levels and observing the fatigue life changes of cables under these load conditions [22]. The key steps of the experiment include observing the behavior of the cable under different loading conditions and revealing the effect of loading on the fatigue performance of the cable material by gradually increasing the load. This paper focuses on the fatigue life, deformation, and other related characteristics of cables and explains the experimental results through statistical and graphical analyses [23]. Although this method can determine the best evaluation solution for a cable design, the test cycle time is long and the evaluation efficiency is low [24]. Leveraging the advantage that finite element simulation can substitute the high cost of experiments, the finite element method proves instrumental in enhancing the efficiency of cable optimization, especially in fatigue life calculations [25]. Suh et al. investigated the impact of stress amplitude and average stress on the fatigue life of multi-strand cables under axial loading using FE-safe fatigue analysis software [26]. Finite element simulation is a numerical method based on mathematical modeling that has become a powerful tool for predicting the fatigue life of structural systems [27]. The technique discretizes a complex structure into finite units, which facilitates the analysis of its behavior under different conditions. The fatigue life of the structure can be predicted and evaluated based on the response of the structure to stresses, deformations, and damage patterns by placing the model in a simulated loading scenario [28]. Combining finite element simulation with fatigue life prediction has far-reaching implications for engineering practice. Engineers can utilize the technique to pre-identify potential vulnerabilities in a structure, thereby facilitating informed decisions on design iterations and material selection [29]. This approach to structural fatigue life analysis is accurate and consistent with the principles of risk reduction and performance optimization in contemporary engineering. Therefore, it is imperative to utilize finite element simulation to eliminate unreasonable factors, thereby improving computational efficiency before conducting relevant tests [30]. The preceding research analysis addresses axial tensile or torsional loads on cables, overlooking the complex nonlinear motion inherent in cable motion [31,32]. Employing a single load, as described above, to simulate the mechanical properties of motion cables introduces deviations from reality and fails to realistically emulate engineering scenarios [13,33]. Equivalent composite model cables respond better with regard to tensile and torsional properties than other cable research methods.

The structure of this paper is as follows. Firstly, ABAQUS 6.14 finite element software is employed to construct a realistic model of motion cables, incorporating judicious simplifications of the cables [34]. Utilizing composite material theory, the equivalent calculation of each material parameter of the cables is conducted to simulate their mechanical properties during the actual movement process [14,35]. Considering the linkage with engine sensors, an analysis is performed to assess the influence of varying tilt angles and fixed-point adjustments on the mechanical properties of the cable. The goal is to determine the optimal working conditions for the motion cable. The fatigue life of the cables is analyzed by the fatigue analysis software FE-safe 6.4 under optimal working conditions, and the effect of cable inclination on fatigue life is studied to provide a basis for motion cables in engineering applications [36,37,38].

## 2. Materials and Methods

### 2.1. Geometrical Parameters

Motion cables can be cycled thousands of times during sensor operation, so modeling the actual cable structure and determining the fatigue life of the motion cables is a critical factor in determining fatigue life. Therefore, the accuracy of real cable structures is of particular importance [39]. As shown in Figure 1a, the cables connected to component B are identified as motion cables, while the remaining cables are classified as static cables. This study is specifically dedicated to analyzing the model of the motion cables connected to component B. It can be seen by observing the real cables that the modeled cables consist of 39 wires arranged in parallel (Figure 1b). Each wire consists of 19 copper wires, each with a diameter of 0.102 mm, enveloped by insulating plastic with a thickness of 0.38 mm (Figure 1c). The 39 wires form a circular cross-section of 6 mm in diameter and are wrapped with 1 mm thick insulating plastic on the outside, giving a total circular cross-section diameter of 8 mm. In the establishment of a solid model, one needs to consider cable tension and torsion at the same time, so solid units are used for modeling. The geometry of the cables is obtained by measuring the path of the physical cables and a mathematical model is generated by sweeping operations in the modeling approach. The material properties of each component of the model are shown in Table 1.

### 2.2. Modelling Method

Before establishing the finite element simulation model, the material parameters of complex cables should be equated according to the mechanical properties of the structure under the premise of ensuring rationality [40,41]. In previous studies, the simulation of cables under tensile or torsional loads in a single direction could directly replicate the actual structure. However, in this study, the cables are subjected to both tensile and torsional loads, leading to a complex modeling process and intricate contact relationships among the wires in the cables. This complexity poses challenges in accurately simulating the real contact process, contributing to a slower computational speed. Therefore, this paper introduces the method of anisotropic composites and implements a simplified treatment of the cables by considering the cables with multiple conductors as unidirectional fiber composites and calculating the material parameters using the composite material theory.

According to the composite material theory, the stiffness coefficient of an equivalent composite material model can be expressed as follows:(1)$$C_{ij}=F_{j}(E_{f,}v_{f},c_{f},E_{m},v_{m},c_{m})$$
where $E_{f}$ is the modulus of elasticity of isotropic fiber, $V_{f}$ is the Poisson’s ratio of isotropic fibers, $E_{m}$ is the modulus of elasticity of the matrix, $V_{m}$ is the Poisson’s ratio of the matrix, $C_{m}$ and $C_{f}$ are the relative volume contents of the fiber and matrix, respectively, $v_{f}$ is the volume of the fiber, and $v_{m}$ is the volume of the matrix. Among them, $E_{f}$, $V_{f}$, $E_{m}$, and $V_{m}$ all are known; $C_{f}$ denotes the volume fraction of the total volume of copper wire in n wires. d is the distance from the center to the inner circle, and D is the distance from the center to the outer circle. In composite materials, the stress–strain relationship can be expressed in terms of a stiffness matrix σ=Cε, The stiffness matrix of the unidirectional fiber composite used in this study can be expressed as follows:(2)$$C=\left[\begin{array}{cccccc} C_{11} & C_{12} & C_{13} & 0 & 0 & 0\\ C_{12} & C_{11} & C_{13} & 0 & 0 & 0\\ C_{13} & C_{13} & C_{33} & 0 & 0 & 0\\ 0 & 0 & 0 & C_{44} & 0 & 0\\ 0 & 0 & 0 & 0 & C_{44} & 0\\ 0 & 0 & 0 & 0 & 0 & C_{66} \end{array}\right]$$

The equations involved in calculating the stiffness factor above are shown below:(3)$$c_{f}=\frac{nr^{2}}{nR^{2}+D^{2}-d^{2}}$$
(4)$$c_{m}=1-c_{f}$$
(5)$$E_{1}=E_{f}c_{f}+E_{m}\left(1-c_{f}\right)$$
(6)$$E_{2}=\frac{E_{f}E_{m}}{c_{m}E_{f}+c_{f}E_{m}}$$
(7)$$v_{21}=c_{m}v_{m}+c_{f}v_{f}$$
(8)$$G_{12}=\frac{G_{m}G_{f}}{G_{m}c_{f}+G_{f}(1-c_{f})}$$

### 2.3. Model Correctness Verification

To verify the accuracy of the above-simplified modeling method, this paper adopts the method of comparing the physical model with the finite element model (Figure 2a,d), which compares in detail the contact situation, morphology, and trajectory similarity at the final angle of the model, focusing on observing the behavior of the model when it reaches the final angle. As shown in Figure 2b,c,e,f, both the physical model and the finite element model maintain no contact with the structural component as they reach the final angle, maintaining a safe distance. This observation indicates that the behaviors of the physical model and the finite element model align. In Figure 2, the final shape of the motion cable is examined, and it is observed that the final shape of the cable is consistent in both the finite element model and the physical model. In addition, the trajectories of the motion cables are compared, and it is found that the two trajectories are similar with an error of less than 5%, as shown in Figure 3. This result further substantiates the accuracy of the numerical model, demonstrating that the model, post-equivalent treatment, can adeptly simulate the motion process of the cables and effectively capture the motion behavior of the actual cables.

### 2.4. Boundary Conditions and Mesh Division

When establishing the finite element simulation model, considering that the simulation analysis only focuses on the stress distribution of the cables, the main structure other than the cables can be set as the rigid body, and the effect of this setup on the simulation results of the cables is very small and significantly saves calculation time. Additionally, neglecting fatigue damage, the simulation results indicate that the cables rotating 170° clockwise and counterclockwise are identical. Therefore, to optimize computation time, only half of the rotational period is considered in this study, focusing solely on analyzing the change in the stress distribution of the cables rotating counterclockwise by 170°. In the constraint set, a load of 170° rotation is applied to the top of the turntable, and the rotation of the cables is driven by the rotation of the turntable, to simulate the whole motion process of the rotation of the cables. In the project, the cables are usually applied to both ends of the ring of the crimp to achieve the purpose of restraining the movement at the end; the fixed end of the constraint is critical to the overall stress situation of the cables. Therefore, this study proposes simulating the forces acting on the cables under various fixed-point positions. To enhance the convenience of engineering calculations, the simulation implements boundary conditions by constraining both displacements and rotations at the boundary. The boundary condition of the cables is set as a friction boundary to make the contact between the fixed end of the cables and the structural member closer to the real contact in the physical model. The end treatment method of the simulated cables proposed in this paper restores the actual stress state of cables in engineering and improves the accuracy of finite element simulation.

The focus of this study is to observe the stress distribution of the cables under the combined effect of torsion and tension, and the complex structure has high requirements regarding the mesh quality. Therefore, for enhanced computational accuracy and efficiency, hexahedral elements are employed to construct the mesh. The mesh generation utilizes the internal axis algorithm, with the cell type designated as C3D8R. The size of the circular cross-section direction is set at 0.3 mm, and seeds are distributed numerically along the axial direction of the line cables. The cables are irregular in shape and the forces are more complicated during the movement. Therefore, the size of the mesh at the boundary conditions is reduced from 0.5 mm to 0.2 mm.

## 3. Results

### 3.1. Effect of Cable Inclination on Stress Distribution in Members

Figure 4a–d demonstrate the stress change process in the motion cables, where the stress concentration appears at the upper end of the cables at the beginning of the movement. At the start of the movement, the top flange is initially rotated, subsequently causing the movement of the cable. The stress distribution of the cable changes as it moves, with the stress concentration area gradually shifting from the top to the middle of the cable. When motion reaches the final state, the stress concentration area is always distributed in the middle of the steel cables’ bending.

Owing to the bending structure of the motion cable, substantial stress variations are induced in the mid-section region. The stress extremes increased with the cable’s rotation angle, reaching the maximum stress at the final moment. The largest stress area is in the middle of the cables, measuring 138 MPa. To explore the trend of the stress maximum region, six reference points are selected along the high-stress region in the middle of the cable at the end moment of the campaign and the magnitudes of the values at different points in the middle of the cable are analyzed, as shown in Figure 4e.

Figure 4f plots the Mises stress–strain curves at six reference points in the high-stress region in the middle of the cables. As the cable moved closer to the bent portion, the stresses at the six reference points increased, with extreme values of 127 MPa being reached at points 5 and 6. The extreme stresses at the other four reference points 4, 3, 2, and 1 (Figure 4e) decreased as they moved away from the bent portion of the cable. At the location of the bent portion of the cable, the area near the center is stressed first, and then, the stresses begin to be distributed in all directions. In summary, the stress extremes are maximized at the curvature maximum in the high-stress region, gradually decreasing as one moves away from this maximum curvature location. Subsequent analysis will focus on the factors influencing the stress distribution in the final state of motion.

The size of the inclination angle of the motion cable is a crucial factor in determining fatigue life; therefore, it is imperative to investigate the impact of the inclination angle on the stress distribution. If the inclination angle is too large, the cable will contact the structural components during the movement. Therefore, only models with an inclination angle of less than 20° are established in this paper, as shown in Figure 5a–f.

The motion process of the cables is simulated through finite element analysis, and the final stress distribution is visualized. For enhanced observation of the stress distribution under various inclination angles, this study selectively displays parts of the cable and turntable model while concealing the midsection of the cylindrical structural member. The stress distribution of the cables remained consistent during the movement, and the stress extremes all appeared in the middle of the cables. A comparison of the stress extremes for the six operating conditions revealed that the extreme stresses increased as the inclination angle increased. In the final state, the stress extreme of the cables is the sum of the stresses under the joint action of tension and torsion. When the inclination angle is increased, the mutual squeezing effect of the intermediate bending segments would be increased and the axial deformation would be increased. This constrains the cable margins existing in the cable structure, leading to a reduction in the radial torsional deformation of the cable. From the values of the modulus of elasticity in the equivalent model, the modulus of elasticity for axial tension is larger than the modulus of elasticity for radial torsion. Therefore, when the axial deformation of the cables increased and the radial deformation decreased, the axial tensile load on the cables increased, and the stress extreme increased from 138.9 MPa to 315.2 MPa (Figure 5g–i). With the movement of the cable, the ultimate stress increased. The pronounced impact of this change along the axial direction is observed in the central region, while the stress extremes increased to a lesser extent in the two end regions. Consequently, the stress in the central region surpassed that in the surrounding area. Therefore, among the six conditions, the cable’s inclination angle of 0° exhibited the lowest tensile load, making it the most favorable condition among the six.

### 3.2. Effect of Cable Fixation Points on Stress Distribution in Members

To investigate the effect of fixed points on the stress distribution of a motion cable, two models were developed. Figure 6a,b, shows two different fixing-point location schemes, in which Scheme B has one more fixing point than Scheme A and the horizontal length is 6 cm longer. As shown in Figure 6c,d, the axial Mises stress distributions are shown for the two fixed-point schemes. The high-stress region of Scheme A is confined to the middle part of the cable bends with a stress extreme of 192 MPa. Conversely, the high-stress region of option B encompasses the middle part of the cables and the top crimp card, reaching a stress extreme of 256 MPa. Upon examining the color representation, it became evident that in the middle portion of the cables, the stress values for fixing point B exceeded those for fixing point A.

Figure 6e,f displays the Mises stress distribution in the radial direction for the two fixing-point scenarios. It reveals that the stress value for fixing point A is 27 MPa, whereas the stress value for model B is 64 MPa. In the high-stress region (Figure 6g), two parallel cells are selected along the axial direction of the motion cables, and the variation in shear stress is analyzed based on the relative angles of the two cells as they are moved to the final angle. In models A and B, the angles between the two parallel units are 13° and 37°, respectively (Figure 6h,i). The existence of tangential stresses during torsion is verified because of the different values of the angles in the two models. In addition, the presence of additional fixation points in the fixed-point B scenario resulted in the application of moments to the entire cables, increasing the angle of torsion for Model B.

Figure 7 displays the energy versus strain curves for fixed point B in fixed-point schemes A and B. The curve for fixed-point scheme A varies linearly, while the curve for fixed-point scheme B varies exponentially. The inclusion of fixed point A results in higher stress in fixed-point scheme B compared to fixed-point scheme A. In fixed-point scheme B, fixed point A induces a moment at fixed point B during rotation and another moment at fixed point B originating from the lower fixed point C. The concentration of these two moments at fixed point B leads to increased stress in this region, resulting in a surge in the strain energy at fixed point B. The initial slopes of energy–strain for fixed-point scheme B are smaller than those for fixed-point scheme A, indicating that too many fixed points constrain the rotation of the cables. When the strain of the cables reaches 0.25, the strain energy slope of fixed-point scheme B increases. At this point, the cable force mode changes, with axial tensile load becoming the main force mode.

### 3.3. Fatigue Simulation and Analysis of Cables under Optimal Working Conditions

After analyzing both the inclination angle and the fixation point position, the optimal configuration is determined to be a structure with a 0° inclination angle and fixation point position A, as illustrated in Figure 8a. Examining the stress cloud of the optimal configuration (Figure 6g), the stress extremes are observed to be distributed in both the middle and lower regions of the cables. This phenomenon is attributed to the rotation of the top flange, generating an axial force transmitted along the cables. This force partly affects the middle of the bend and partly reaches the fixed point at the lower end of the cables.

The static analysis results mentioned above are imported into the Fe-safe fatigue analysis software, and a fatigue simulation cloud diagram for the motion cables is generated (Figure 8a). The areas of low peripheral fatigue in the cables are depicted in three parts, located at the top, middle, and bottom. To examine strain changes in these three parts during the loading cycle, three tracking points are defined within the potential fatigue region of the cables, as illustrated in Figure 8a. These tracking points represent the average strain values in each of the three regions. Extracting the fatigue life and strain changed curves of the three tracking points (Figure 8b), it is observed that when the overall strain of the cables reached 0.05, the fatigue life of the first tracking point is 45,163, the fatigue life of the second tracking point is 37,256, and the fatigue life of the third tracking point is 42,563. The data reveal that the low-week fatigue region aligns with the stress concentration region, suggesting that the high-stress region is prone to fatigue damage. The fatigue life of the middle section is significantly lower than that of the top and bottom, indicating that the primary location of fatigue damage in the cables is the middle section.

## 4. Conclusions

In this paper, the mechanical properties, stress distribution, and fatigue life of motion cables are investigated in aero-engine sensors under complex motion conditions. Employing the composite material theory, an equivalent cable model is developed and the forces acting on the cable were analyzed using the finite element method. The accuracy of the equivalent model is verified by comparing it with experimental results. Our analysis focuses on understanding the effects of the inclination angle and fixation point on the stress distribution of the cables, allowing us to determine optimal operating conditions to minimize extreme stresses during motion. Furthermore, the impact of the tilt angle of the tension cables on the stress distribution of the member is explored, revealing its significant effect on the final stress distribution of the model. Simulations involving six different tilt conditions concluded that a 0° tilt angle is the optimal condition, resulting in a stress extreme of 139 MPa. Additionally, the analysis considers the effect of different fixation points on the stress distribution in the members and determines that Model A is superior to the other model. Therefore, maintaining the cable inclination angle at 0° and minimizing the number of fixing points is recommended to ensure the safety and stability of the cables. Furthermore, fatigue analysis based on the optimum conditions indicates that the high-stress zone and the short-fatigue life zone are situated in the middle of the cables. The ultimate stress in this zone is 138 MPa, corresponding to 10^7.706^ loading cycles.

## Figures and Tables

**Figure 1 sensors-24-01109-f001:**
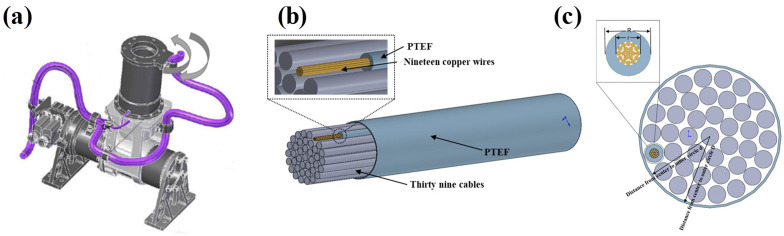
Internal composition of cables: (**a**) overall schematic of the drive mechanism; (**b**) overall arrangement of the 39 wires in cables; (**c**) arrangement of the copper wires in each wire.

**Figure 2 sensors-24-01109-f002:**
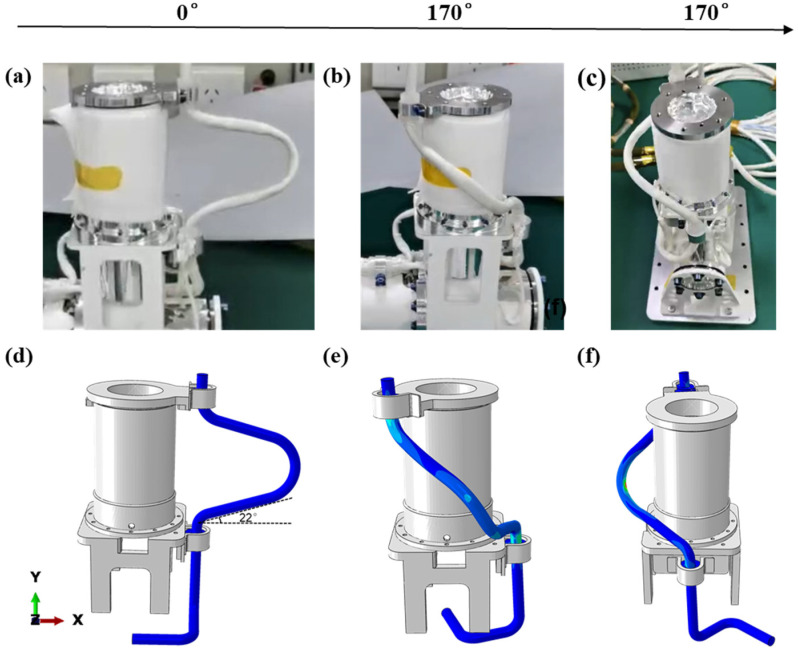
Comparison of poses of the physical model and the simulation model at the initial and final angles: (**a**) front view of the physical model at the initial angle; (**b**) front view of the physical model at the final angle; (**c**) side view of the physical model at the final angle; (**d**) front view of the simulation model at the initial angle; (**e**) front view of the simulation model at the final angle; (**f**) side view of the simulation model at the final angle.

**Figure 3 sensors-24-01109-f003:**
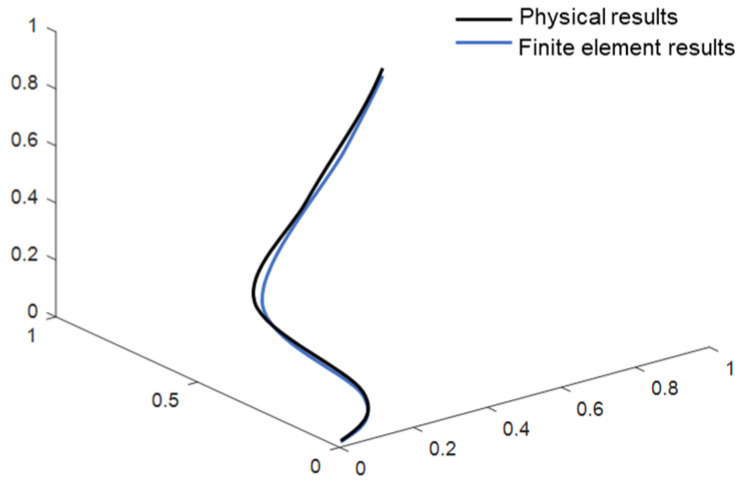
Comparison between physical results and finite element results.

**Figure 4 sensors-24-01109-f004:**
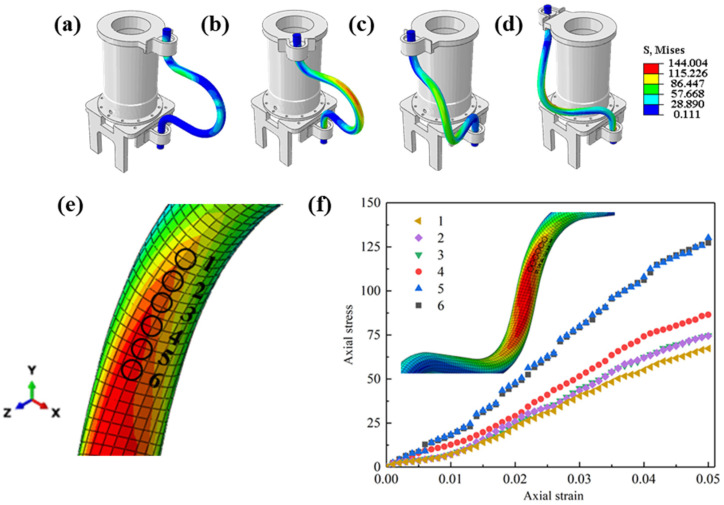
Mises stress distribution, the middle part of cable bend, and the position of the probes as well as the energy–strain curves at different angles of rotation: (**a**) 5°; (**b**) 42.5°; (**c**) 85°; (**d**) 170°; (**e**) position of the applied probes; (**f**) energy–strain curves at the six probe points.

**Figure 5 sensors-24-01109-f005:**
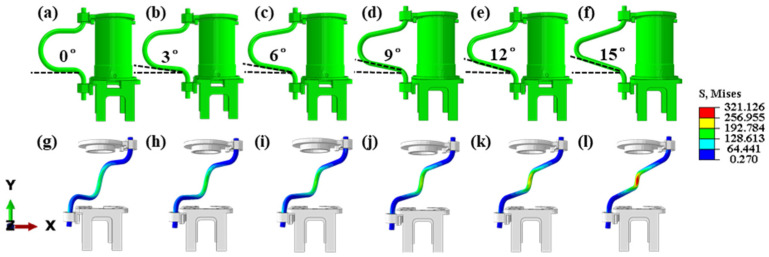
Cables’ model and stress distribution for different inclination angles: (**a**) 0°; (**b**) 3°; (**c**) 6°; (**d**) 9°; (**e**) 12°; (**f**) 15°; (**g**) 0°; (**h**) 3°; (**i**) 6°; (**j**) 9°; (**k**) 12°; (**l**) 15°.

**Figure 6 sensors-24-01109-f006:**
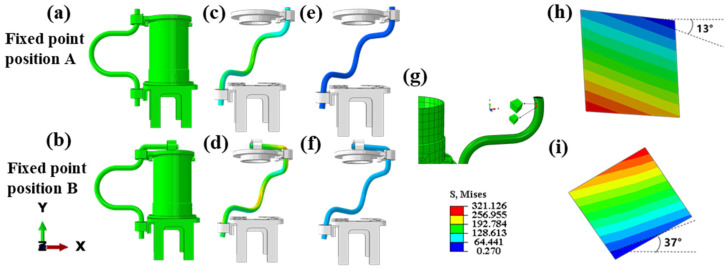
Schematic of working conditions and stress distribution for different fixed points. (**a**) Schematic of fixed-point scheme A; (**b**) schematic of fixed-point scheme B; (**c**) stress distribution of fixed-point scheme A on S11; (**d**) stress distribution of fixed-point scheme B on S11; (**e**) stress distribution of fixed-point scheme A on S23; (**f**) stress distribution of fixed-point scheme B on S23 torsion angle of neighboring units in fixed-point scheme A; (**g**) position of two neighboring units in the initial state; (**h**) torsion angle of neighboring units in fixed-point scheme A; (**i**) torsion angle of neighboring units in fixed-point scheme B.

**Figure 7 sensors-24-01109-f007:**
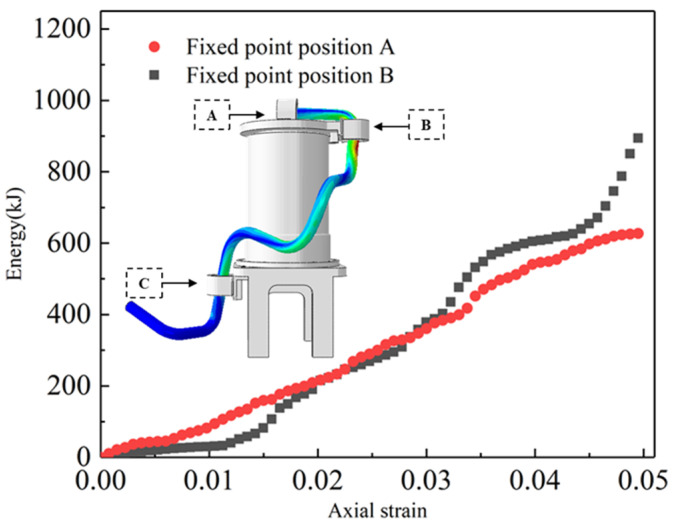
Strain versus strain energy curves in models A and B.

**Figure 8 sensors-24-01109-f008:**
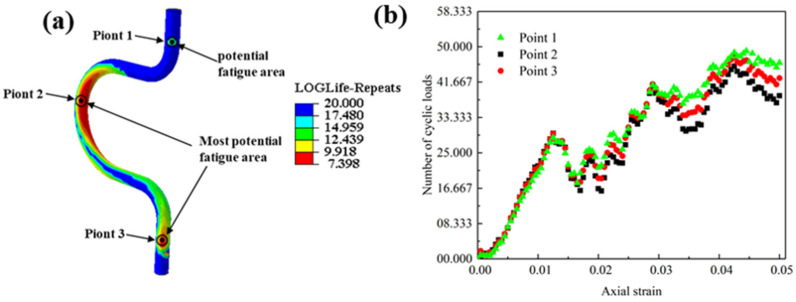
Number of cyclic loads for optimum conditions: (**a**) fatigue life distribution; (**b**) strain vs. fatigue life curves for three tracking points.

**Table 1 sensors-24-01109-t001:** Material properties.

Component	Materials	Density (kg/m^3^)	Young’s Modulus (GPa)	Poisson’s Ratio	Yield Strength (MPa)	Tensile Strength (MPa)
Main part	Aluminum alloy	4510	117.2	0.33	860	900
Cables	Copper	8890	110	0.326	369	448
Cables shell	Polytetrafluoroethylene	1750	0.6	0.4	45	34.5
Ring of crimp	Aluminum	2670	72	0.3	80	175

## Data Availability

The data presented in this study are available on request from the corresponding author.

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
