# Peer review of "Mechanical Properties and Fatigue Life Analysis of Motion Cables in Sensors under Cyclic Loading"

_sensors, 2024, doi:10.3390/s24041109_

Round 1

Reviewer 1 Report

Comments and Suggestions for Authors

The work is meaningful, but the introduction section is relatively simple and cannot provide readers with a clear understanding of the significance of this article. It is recommended that the author supplement this section and resubmit it. The author did not provide a good introduction to the research background and significance, and did not compare the performance improvements studied. When introducing the review, the article only mentioned that it was through experimental testing to determine the factors that reduce cables fatigue life, but did not elaborate on the specific methods. Even if the author mentions finite element simulation, only two people's studies are listed, which is not enough to indicate that your research is meaningful.

Comments on the Quality of English Language

Need improvement, preferably after polishing

Reviewer 2 Report

Comments and Suggestions for Authors

Dear authors, thank you for your work.

Best regards,

reviewer

Reviewer 3 Report

Comments and Suggestions for Authors

Although the authors have selected an interesting area for this research, the article needs serious revisions in the light of the following observations:

1. The authors extensively mention reliability without elaboring on the concept and without distinguishing it from reliability.

2. The authors should explain the technique used to predict reliability and how strong is this technique.\

3. What are the factors and parameters used to model the reliability.

4. The language usage is far below the expectation for a journal article. There are several language usage errors. Overall, a formal technical reporting format should be strictly adhered to and use of first person statements should be avoied. Thorough proofreading is recommebded to be done by an English speaker/professor/instructor. 

Comments on the Quality of English Language

The language usage is far below the expectation for a journal article. There are several language usage errors. Overall, a formal technical reporting format should be strictly adhered to and use of first person statements should be avoied. Thorough proofreading is recommebded to be done by an English speaker/professor/instructor. 

Round 2

Reviewer 1 Report

Comments and Suggestions for Authors

The author has completed all the revisions and recommends the article for acceptance.

Reviewer 3 Report

Comments and Suggestions for Authors

The authors are advised to address the review points in the light of the definition of reliability e.g. the American Society for Quality "Reliability has sometimes been classified as "how quality changes over time." The difference between quality and reliability is that quality shows how well an object performs its proper function, while reliability shows how well this object maintains its original level of quality over time, through various conditions." Also the authors should present the reliability model they used for FEM simulations. The facors affecting reliability should feature in this model.

Comments on the Quality of English Language

There are still language usage errors. Overall, a formal technical reporting format should be strictly adhered to and use of first person statements should be avoied. Thorough proofreading is recommebded to be done by an English speaker/professor/instructor. 
